# Efficacy of pharmacological and non-pharmacological therapy on chronic cancer pain intensity of adults with cancer: A network meta-analysis protocol

Wenhao Su[1], Xueling Li[2], Yanru Wang ![ORCID][3]*

1 Nursing Department, The Quzhou Affiliated of Wenzhou Medical University, Quzhou People's Hospital, Quzhou, Zhejiang, China, 2 School of Nursing, Hainan Vocational University of Science and Technology, Haikou, Hainan, China, 3 School of Nursing, Zhejiang Chinese Medical University, Hangzhou, Zhejiang, China

* Wangyanru001@outlook.com

## Abstract

### Background

Chronic cancer pain is very common symptom in cancer patients, but this issue has not been satisfactorily resolved by the conventional three-step analgesic therapy. There are multiple non-pharmacological interventions for managing chronic cancer pain, but we haven't reached a consensus on which non pharmacological treatment is the best and these treatments are lack of high-quality evidence. In order to identify the most effective non-pharmaceutical therapy alternatives and investigate further possible medication interventions, this study will use network meta-analysis to assess the therapeutic effects of pharmacological and non-pharmacological treatments on chronic cancer pain patients and support clinical decision-making by prioritizing therapies according to the most valuable clinical outcomes for these patients.

### Methods and analysis

We will carry out a systematic search of published randomized controlled trials (group, crossover, and parallel) in the PubMed, Web of Science, Cochrane Library, MEDLINE, Embase, and CINAHL databases, without language or date restrictions, in accordance with the PRISMA for Network Meta-Analyses (PRISMA-NMA) guidelines. Included studies must evaluate the effects of pharmacological and non-pharmacological treatments in patients with chronic cancer pain. Adult chronic cancer pain patients (≥ 18 years old) receiving pharmacological or non-pharmacological treatment will be our target participants. Our primary outcomes will be pain intensity, total effective rate of treatment, onset time, and quality of Life (QoL); Adverse reaction will be our secondary outcome. We'll utilize the mean difference (MD) for

**Data availability statement:** All relevant data are within the manuscript and its Supporting Information files.

**Funding:** The author(s) received no specific funding for this work.

**Competing interests:** The authors have declared that no competing interests exist.

continuous variables, the odds ratio (OR) for binary variables, and the 95% confidence interval (CI) for interval estimates. The Cochrane Bias Risk Tool (RoB2.0) will be used to assess the bias risk of every RCT trial included in NMA. We will use Review Manager 5.3 software to conduct heterogeneity testing and meta-analysis. The network meta-analysis will be performed by ADDIS1.16.8 software. The Confidence in Network Meta-analysis (CINeMA) framework will be used to evaluate the level of confidence in the NMA results. Besides, we will use SUCRA for ranking the network meta-analysis results, and we will also apply normalized entropy to verify the accuracy of the SUCRA ranking outcomes.

## Discussion

This network meta-analysis will compare the efficacy of pharmacological versus non-pharmacological treatments for pain intensity in chronic cancer pain patients. The final analysis results may be significantly heterogeneous, because the population with cancerous pain suffers from different types of cancers. Owing to the databases primary reliance on our listed databases for inclusion, potentially valuable research will be overlooked.

## Registration

This study has been registered in the PROSPERO database (CRD42024505214)

## Introduction

Chronic cancer pain, arising from primary or metastatic tumors or their treatments, is a chronic condition lasting over three months [1]. It affects up to 70% of patients with advanced cancer, with one-third experiencing severe pain [2,3]. This significantly impacts the quality of life for patients, their families, and caregivers [4] and imposes a substantial financial burden on healthcare systems [5]. Advances in diagnostic and therapeutic technologies have increased patient longevity, and effective pain management may positively influence survival in several cancers [6].

Alleviating chronic cancer pain has become a crucial clinical issue. The three-step pain ladder, widely used in medical facilities, can alleviate pain in over 70% of cancer patients [6–8]. The ladder consists of: (1) oral administration; (2) administering on time; (3) administering in steps; and (4) individualizing medication dosages. Based on the patient's pain level and underlying cause, physicians choose appropriate analgesics. For first-grade (mild) pain, NSAIDs or acetaminophen with or without adjuvants are used. For second-grade (moderate) pain, weak opioids (hydrocodone, codeine, tramadol) are used with or without non-opioid analgesics and adjuvants. For third-grade (severe) pain, potent opioids (morphine, methadone, fentanyl, oxycodone, buprenorphine, tapentadol) are administered with or without non-opioid analgesics and adjuvants [8,9]. Additionally, the dosage of analgesics should be progressively increased from weak to strong [10].

However, the adverse effects of opioid medications, such as opioid-induced neurotoxicity [11], constipation [12], nausea [13], and respiratory depression [14], present a significant challenge in managing chronic cancer pain [15]. Opioids may also lower the immune system's efficacy, compromising anti-cancer therapies [16]. They can suppress T lymphocytes and natural killer cells, impairing immune function and increasing infection rates [16]. As cancer patients, particularly those with advanced cancer, require higher analgesic dosages and have lower health-related quality of life [17], non-pharmacological therapies have been recommended by the National Comprehensive Cancer Network (NCCN) clinical practice guidelines for adult chronic cancer pain [18].

Despite guidelines recommending opioid medications [2], inadequate treatment is still widespread, with 32% to 82.3% of patients not receiving adequate pain management [19]. This is due to factors such as difficult access to medication for at-home care, overly cautious prescribing by clinicians [20], and strict regulations on analgesic use in some countries. The focus of chronic cancer pain treatment guidelines primarily on pharmacological therapy often neglects the specificity of elderly patients with advanced cancer, who may experience adverse effects from comorbities and multiple medications.

Chronic cancer pain is very common symptom in cancer patients, but this issue has not been satisfactorily resolved by the conventional three-step analgesic therapy. There are multiple non-pharmacological interventions for managing chronic cancer pain, but we haven't reached a consensus on which non pharmacological treatment is the best and these treatments are lack of high-quality evidence.

Conventional meta-analyses, limited to comparing one or two interventions, may not provide all the data required for clinical decision-making. Network meta-analysis allows for the evaluation of multiple interventions and the examination of both direct and indirect evidence, offering a more accurate estimation when evidence is limited to a few substandard studies [21,22]. Therefore, our network meta-analysis aims to examine the effects of pharmacological and non-pharmacological therapies on chronic cancer pain patients, identifying the most effective non-pharmacological interventions and other potentially effective pharmacological treatments.

## Methods and analysis

The protocol was registered to PROSPERO database on February 12, 2024 (CRD42024505214). We will structure the material of the systematic review and NMA using the PRISMA-NMA extension statement [23]. This meta-analysis will be carried out in accordance with the Preferred Reporting Items for Systematic Reviews and Meta-Analyses (PRISMA) statement. We have completed the PRISMA-NMA(S1 File) and PRISMA-P (S2 File). A summary of eligibility criteria for Population, Intervention, Comparison, Outcome, and Study design (PICOS) is accessible in S3 File.

### Eligibility criteria

In this network meta-analysis, we will include participants diagnosed with chronic cancer pain, according to the criteria defined by the International Association for the Study of Pain (IASP) [1]. While it is generally accepted that chronic pain is defined as pain lasting longer than three months, the precise concept of time is not stressed in cancer pain because cancer patients may experience rapid disease progression and shortened survival times. Therefore, we will define chronic cancer pain is a kind of chronic pain caused by the primary cancer itself or metastases (chronic cancer pain) or its treatment (chronic post-cancer treatment pain) [1].

We will include randomized controlled trials (RCTs) that feature a control group for efficacy comparison. The control group can receive a placebo, conventional treatment, or any intervention different from the primary treatment method. If a study lacks a control group, it must demonstrate that the target intervention has higher efficacy compared to other interventions applied in the sample.

The studies must employ a randomized controlled trial design to assess the effectiveness of both pharmacological and non-pharmacological treatments for chronic cancer pain patients. Non-pharmacological interventions must not impose any pharmacological effects on participants.

Participants must be adult males or females, at least 18 years old, experiencing pain caused by cancer. Both race and gender will be considered independent variables in the research.

## Exclusion criteria

We will exclude randomized controlled trials (RCTs) where participants primary health condition is not chronic cancer pain or where the studies do not retain complete study data. Studies will be excluded if they do not meet the following criteria: they do not have a control group that received a different treatment from the one being studied; they do not demonstrate substantial efficacy of the target therapy compared to the other therapies included in the study; they include individuals who are awaiting a biopsy or a diagnosis; they conduct pain research specifically focused on diseases other than cancer; they have too broad inclusion criteria that include participants who do not meet the requirements of this study.

## Interventions

In this study, non-pharmacological intervention will refer to the implementation of management measures that do not rely on the administration of nutritional supplements or medicines. We have established a predetermined set of intervention measures (S4 File) derived from previously published RCTs, systematic reviews, and clinical practice guidelines. The intervention measures we will be interested in are the following: pharmacological therapies [8] including nonsteroidal anti-inflammatory drugs (NSAIDs) such as aspirin, acetaminophen, diclofenac, indomethacin, quinolone, ibuprofen, naproxen, flurbiprofen esters, piroxicam, meloxicam, paclitaxel, and celecoxib; weak opioids such as codeine, oxycodone, and tramadol; and strong opioids such as morphine hydrochloride, morphine sulfate, remifentanil, pethidine, buprenorphine, and oxycodone. Non-pharmacological therapies include acupuncture [24–27], cognitive behavioral strategy [28], mindfulness and meditation [29–31], psychological therapy [32], spiritual intervention [33], electrotherapy [34,35], hypnosis [36,37], massage therapy [24,38,39], music therapy [28,40,41], telephone-based interventions [42], yoga, Tai Chi, and Qigong [28,43], and complementary therapies [44]. Each intervention mentioned above controls for completely different confounding factors, so we will create three different control groups that include "placebo & sham", "no intervention & waiting list", and "standard drug treatment". To ensure that various interventions do not violate the assumption of transitivity for the network meta-analysis (NMA), we will define the "standard drug treatment" as following the three-step pain ladder characterized by nonsteroidal anti-inflammatory drugs, weak opioids, and strong opioids. In this network meta-analysis, comparisons of interventions will be made with each other. All control groups in the studies we will include will be assigned to these three groups.

## Outcomes

The results will be categorized into primary and secondary outcomes. We will include studies that assessed at least one of the following indicators:

**Primary outcomes.**

**Pain intensity**   We will prioritize obtaining data from a single-dimensional pain scale. We will prioritize extracting numerical rating scale (NRS) data. If the studies included in the analysis do not utilize NRS, we will include the data in the following order: visual analog scale (VAS), verbal rating scale (VRS), verbal descriptor scale (VDS), Wong Baker faces pain scale revision (FPS-R), and Mankoski pain scale (MPS). The examination of multidimensional pain tools will only be considered if there is no alternative tool available for a single dimensional pain scale. We are going to use the multi-dimensional assessment tool in following orders if there is no single-dimensional pain scale being used in the included trials: brief pain inventory (BPI), short-form McGill pain questionnaire (SF/MPQ), and global pain scale (GPS). We will prioritize extracting the average pain, mildest pain, most severe pain, and pain data during rest and activity over the past 24 hours. The pain intensity assessment tool will be described in the trial's characteristics.

**Total effective rate of treatment**   If the study does not clearly disclose the total effective rate of treatment data, we will calculate them by hand. To calculate the total effective rate of treatment, we will divide the total number of effective treatments in the study by the total number of interventions. A effective treatment is defined as a reduction of at least 40% in patient pain intensity or frequency before and after treatment. The total effective rate of the treatment assessment tool will be described in the trial's characteristics.

**Onset of analgesia**   The onset of analgesia will be identified by assessing the time it takes for patients to experience substantial pain alleviation or complete elimination of pain following analgesic treatment.

**Quality of life**   We will prioritize collecting data from the European Organization for Research and Treatment of Cancer Quality of Life Questionnaire 30-Item Core Instrument (EORTC QLQ-C30). If the EORTC QLQ-C30 cannot be obtained from the studies included, we will collect data from the Functional Assessment of Cancer Therapy General Scale (FACT-G). We will also include other quality of life assessment tools, and quality of life assessment tool will be listed in the trial's characteristics as well.

**Secondary outcome.**

**Adverse reactions**   As secondary outcome indicators, we will include treatment reactions such as nausea, vomiting, constipation, respiratory depression, fluctuations in blood pressure, and death. Since objective techniques cannot accurately measure subjective symptoms, we will exclude subjective symptom measurement instruments from the report. For measurable adverse reactions such as blood pressure, we will report them in the adverse reactions section.

**Search design**   Study design incorporates parallel groups, cross-over, and cluster randomized controlled trials. In crossover designs, only data from the initial trial period will be retrieved to exclude any potential carryover effects. All included studies must have a minimum of two different intervention measures.

**Electronic search strategy**   We will establish an electronic literature search strategy document, which will be implemented in PubMed, Web of Science, the Cochrane Library, MEDLINE, Embase, and CINAHL without any language or date restrictions. Additionally, we will conduct a thorough investigation of grey literature and make efforts to contact the authors of articles that lack necessary information. We will also manually search for published systematic reviews on this topic and include any relevant full-text articles from their reference lists.

**Search strategy**   The search method was guided by the PICOS criteria and will consist of three sets of terms pertaining to (1) randomized trials, (2) chronic cancer pain, and (3) therapies. To enhance search results, we will utilize the Medical Subject Headings (MeSH), together with their synonyms and free-text keywords found in titles and abstracts. Here are the MeSH terms that we are going to search in PubMed: "Cancer Pain", "Randomized Controlled Trial", "Yoga, Music Therapy", etc. The detailed literature retrieval approach can be found in the S5 File.

## Study selection and data extraction

After the literature search, the results will be exported to the reference manager Endnote X9 software, where any duplicate studies will be removed. Two independent reviewers, W.H. and X.L., will then examine the titles and abstracts and evaluate any potential full-text material. Studies that meet the eligibility criteria will be included in the review. If necessary, the authors will be contacted via email for clarification, with emails sent every Monday and Friday for up to a month. If the authors fail to respond, the study will be excluded, and the reasons will be documented in the PRISMA flowchart. Any disagreements between the two reviewers will be resolved by a third reviewer, Y.R. We will use Excel 2023 to record the data of the included studies and Review Manager 5.3 to conduct a comprehensive analysis of the extracted data. The PRISMA flowchart is available in the attached file (S1_Fig).

For data extraction, two independent reviewers (W.H. and X.L.) will use a pre-designed sheet, with any disagreements resolved by Y.R. The extracted data will include information on study methods, such as country, first author, study design (parallel group, crossover, or cluster-randomized control trial), number of research centers, and year of publication. Participant details will cover sample size, average age, gender, duration of illness, baseline pain score, comorbities, medication

use, diagnostic criteria, inclusion and exclusion criteria, randomization, medication discontinuation in each group, and whether the data were analyzed according to the intention-to-treat principle. For the intervention, we will record the name, dosage, intervenor, duration, frequency, follow-up period, and main characteristics. Results will include primary outcomes such as pain intensity, total effective rate of treatment, onset of analgesia, and quality of life, as well as secondary outcomes like adverse reactions. We will also note the measurement tools and times, along with the language of publication and any contact with authors for further information.

For continuous variables such as pain intensity, total effective rate of therapy, onset time of pain alleviation, and quality of life, we will collect the mean, standard deviation (SD), and sample size for each group. For discrete data, we will calculate the variance, confidence interval, and sample size for each group. Data will be collected at three time points: 24 hours, 24 hours to 4 weeks, and 4–12 weeks after the intervention. If multiple time points are reported, the time point closest to the end of the intervention will be prioritized. If intervention scores are not available, we will extract the mean changes from baseline and their SDs. Additionally, we will record the number of individuals who completed therapy, those who abruptly discontinued treatment, and those who responded effectively to treatment. If the number of withdrawals is not reported, we will use the figures for the number of effective treatments and the total number of participants to compute the overall treatment efficacy rate.

In cases where the SD is not provided, we will estimate it using available data such as the range value, interquartile range, p-value, standard error, or confidence interval. If the necessary data cannot be inferred, we will contact the study authors via email, sending up to three emails with a seven-day interval between each. If the authors do not respond, the study will be included in the review but excluded from the quantitative analysis. For crossover randomized controlled trials (RCTs), we will only include data from the initial randomization period.

### Risk-of-bias in the included studies

Two independent trained reviewers (W.H. and X.L.) will evaluate the included trials' methodological quality. The following 6 domains will be assessed: (1) bias resulting from the randomization process,(production of randomized allocation sequences, use of allocation concealment) (2) bias due to deviations from intended interventions,(blinding of patients and treating physicians) (3) bias due to missing outcome data,(blinding of the outcome indicator evaluator) (4) bias in the measurement of the outcome,(completeness of outcome data, number and causes of dropouts in various intervention groups, differences in subjects with and without outcome data, handling of missing values) and (5) bias in the selection of the reported result (selective reporting of outcome measures). (6) Other biases: funding sources, and baseline balance between the intervention group and control group.

In cases of disagreement between the two reviewers, an authoritative third reviewer (Y.R.) will make the final decision.

The procedures, steps, team, and tasks of the network meta-analysis review to which this protocol refers are shown in S2_Fig.

S2_Fig Picture chart of the protocol for the network meta-analysis steps. Created at 2024 Biorender, online software, Hangzhou, China.

### Data synthesis and analysis

We will use ADDIS v1.16.8 software for meta-analysis and heterogeneity assessment. For dichotomous data, we will analyze using odds ratios (OR); for continuous variables, we will use mean differences (MD). A 95% confidence interval (CI) will be used to estimate the range of values. The Q test will be used to detect heterogeneity among studies, and the $I^2$ statistic will be used to assess the degree of heterogeneity. If $I^2 \geq 50\%$ and $p \leq 0.10$, a random-effects model will be used for analysis. Conversely, if $I^2 < 50\%$ and $p > 0.10$, a fixed-effects model will be used.

We will use the GeMTC package in R software to conduct Bayesian network meta-analysis (NMA) of the selected randomized controlled trials (RCTs). The node-splitting model (NM) will be used to check the consistency of the

evidence network. If the observed differences are not statistically significant (P > 0.05), a consistency model (CM) will be used for the NMA; if the differences are statistically significant (P < 0.05), an inconsistency model (IM) will be used. We will run 50,000 simulation iterations and 10 burn-in iterations, and use the potential scale reduction factor (PSRF) to assess model convergence. When the PSRF value is close to or equal to 1, it indicates strong model convergence and reliable consistency model results. The surface under the cumulative ranking curve (SUCRA) will be calculated to determine the effectiveness of each intervention. SUCRA values range from 0 to 1, with a value of 1 indicating complete effectiveness and a value of 0 indicating no effectiveness. Interventions will be ranked based on their SUCRA values.

For dichotomous outcomes, we will use odds ratios (OR) to measure the relative effects between different interventions. For continuous outcomes, we will use standardized mean differences (SMD). We will use 95% confidence intervals (CI) to represent the estimated range of effect sizes. If the 95% CI does not include the null value (e.g., 1 for OR, 0 for SMD), the difference will be considered statistically significant. Additionally, we will use a p-value <0.05 as the criterion for statistical significance.

We will use the CINeMA framework to assess the credibility of the NMA results. CINeMA extends the GRADE (Grading of Recommendations, Assessment, Development, and Evaluations) approach by considering six domains to evaluate the certainty of evidence: within-study bias, reporting bias, indirectness, imprecision, heterogeneity, and incoherence. Evaluating the certainty of evidence through these methods will enhance the transparency, reproducibility, and reliability of the results. Two authors (W.H. and X.L.) will independently judge the quality of evidence (high, moderate, low, or very low), and any disagreements will be resolved by a third reviewer (Y.R.).

### Risk of bias across studies

To assess the presence of publication bias or small sample effects, a corrective funnel plot will be created for outcome measures having a literature volume of 10 or above.

We will use the CINeMA framework to assess the credibility of the NMA results. CINeMA extends the GRADE (Grading of Recommendations, Assessment, Development, and Evaluations) approach by considering six domains to evaluate the certainty of evidence: within-study bias, reporting bias, indirectness, imprecision, heterogeneity, and incoherence. Evaluating the certainty of evidence through these methods will enhance the transparency, reproducibility, and reliability of the results. Two authors (W.H. and X.L.) will independently judge the quality of evidence (high, moderate, low, or very low), and any disagreements will be resolved by a third reviewer (Y.R.).

### Publication bias assessment

We will use a calibrated funnel plot analysis to assess publication bias. If the funnel plot is asymmetrical, we will further investigate potential sources of publication bias.

## Discussion

Controlling and treating chronic cancer pain, particularly in late-stage patients, remains a significant health challenge. The standard three-step analgesic ladder often struggles to effectively address this issue. Our objective is to evaluate published RCT studies on chronic cancer pain interventions, and we will adhere to the PRISMA-P and PRISMA-NMA guidelines in our study composition. The CINeMA framework will be utilized to assess the credibility of our analysis, considering factors such as internal bias, indirectness, and uncertainty [45]. We will use the ROB 2.0 tool to evaluate various aspects of internal bias in the studies, including intervention bias, missing outcome data, and randomization processes [46], and incorporate non-pharmacological interventions into our NMA.

Initial mild chronic cancer pain is typically treated with NSAIDs. As the disease progresses, high doses of potent opioid medications may be required. Prolonged use of these drugs can lead to severe adverse effects. Therefore, additional

effective intervention strategies are needed beyond pharmacological treatments. To date, only two NMA studies [47,48] have compared non-pharmacological interventions in chronic cancer pain management, and only one NMA studies [49] have compared pharmacological interventions. Acupuncture-point stimulation has the potential to alleviate moderate to severe cancer pain as well as minimize the negative effects of opioids, including nausea, vomiting, and constipation [47]. The three-step pain ladder in conjunction with manual acupuncture is the most effective in terms of clinical efficacy and functional activity status improvement for patients. When considering the length of analgesic and analgesic duration, combined acupoint moxibustion produces the best results [48]. After analyzing 81 randomized controlled trials, there are considerable variations in efficacy between existing regimens for chronic cancer pain. They found that certain non-opioid analgesics and nonsteroidal anti-inflammatory drugs can serve as effectively dezocine, and the NSAID diclofenac were the most effective individual treatments [49]. However, these studies did not encompass all available interventions, limiting their effect size and level of evidence. Our research aims to identify measures that can alleviate severe pain, surpassing the traditional three-step analgesic ladder, to improve patients' quality of life and pain control. as opioids in managing chronic cancer pain. And in terms of pain intensity, they discovered that no pharmaceutical class significantly improved pain intensity compared to placebo, but the non-opioid analgesic ziconotide, the opioid

Our NMA has several advantages. We will conduct a comprehensive search of existing pharmacological and non-pharmacological interventions for chronic cancer pain and rank these treatments using SUCRA. Additionally, we will employ normalized entropy to verify the SUCRA ranking results, enhancing the reliability of our findings.

Currently, there is a lack of research explicitly comparing various chronic cancer pain interventions. We anticipate that our NMA will provide valuable treatment recommendations to healthcare professionals, researchers, and patients, assisting clinicians in making more informed decisions.

## Supporting information

**S1 Fig. PRISMA flow diagram describing the selection process of studies.**
(TIF)

**S2 Fig. Picture chart of the protocol for the network meta-analysis steps.**
(TIF)

**S1 File. PRISMA-NAM.**
(PDF)

**S2 File. PRISMA-P.**
(PDF)

**S3 File. Summary of PICOS eligibility criteria.**
(PDF)

**S4 File. Intervention node.**
(PDF)

**S5 File. Search strategy.**
(PDF)

## Acknowledgements

The authors would like to acknowledge the following contributor: Shumin Liu, a nursing professor, for her great help in forming the search syntax.

## Author contributions

**Conceptualization:** Wenhao Su, Yanru Wang.

**Methodology:** Wenhao Su, Yanru Wang.

**Supervision:** Yanru Wang.

**Writing – original draft:** Wenhao Su, Xueling Li.

**Writing – review & editing:** Wenhao Su, Xueling Li, Yanru Wang.

**Writing – revising & editing:** Wenhao Su, Xueling Li, Yanru Wang.

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
