## [Decision Letter · Decision Letter 0]

20 May 2024

Dear Dr. Yanru,

We look forward to receiving your revised manuscript.

Kind regards,

Marcus Tolentino Silva, Ph.D.

Academic Editor

PLOS ONE

Journal Requirements:

3. We notice that your supplementary figures and tables are uploaded with the file type 'Other'. Please amend the file type to 'Supporting Information'. Please ensure that each Supporting Information file has a legend listed in the manuscript after the references list.

4. Please update your submission to use the PLOS LaTeX template. The template and more information on our requirements for LaTeX submissions can be found at http://journals.plos.org/plosone/s/latex.

Additional Editor Comments:

**ACADEMIC EDITOR:**General :

1. Ensure all actions are written in future tense.

Abstract :

3. Rewrite the abstract to follow a clear and logical sequence, using PRISMA-NMA guidelines.

4. Include all study objectives, such as the total effective rate of treatment.

5. Briefly describe the statistical model and data analysis procedures.

Introduction

8. Revise the discussion of pharmacological dependence and adverse effects to ensure clarity and logical flow.

9. Correct the term to "network meta-analysis."

Methods

12. Include a summary of the eligibility criteria (PICOTS) within the manuscript.

13. Define the diagnostic criteria for the sample.

14. Describe the types of randomized clinical trials to be included.

15. Clarify that subjects with pain not related to cancer will be excluded.

16. Clearly define the pre-defined intervention sets based on published guidelines and systematic reviews.

17. Create separate control groups for "placebo+sham," "no intervention+waiting list," and "standard drug treatment" to ensure transitivity.

18. Define criteria for the "standard drug treatment" group to maintain consistency.

19. Clearly state primary and secondary outcomes in separate paragraphs.

20. Develop a detailed description of data extraction variables for methods, participants, interventions, and outcomes.

21. Address the assumption of transitivity and coherence.

Discussion :

23. Reiterate that the protocol was written following the PRISMA-P and PRISMA-NMA.

Reviewers' comments:

Reviewer's Responses to Questions

**Comments to the Author**

1. Does the manuscript provide a valid rationale for the proposed study, with clearly identified and justified research questions?

Reviewer #1: Partly

Reviewer #2: Partly

2. Is the protocol technically sound and planned in a manner that will lead to a meaningful outcome and allow testing the stated hypotheses?

Reviewer #1: Partly

Reviewer #2: Partly

3. Is the methodology feasible and described in sufficient detail to allow the work to be replicable?

Reviewer #1: No

Reviewer #2: No

4. Have the authors described where all data underlying the findings will be made available when the study is complete?

Reviewer #1: No

Reviewer #2: Yes

5. Is the manuscript presented in an intelligible fashion and written in standard English?

Reviewer #1: No

Reviewer #2: No

You may also provide optional suggestions and comments to authors that they might find helpful in planning their study.

Reviewer #1: Thank you for the opportunity to review this manuscript.

The manuscript is a network meta-analysis protocol assessing the efficacy of pharmacological and non-pharmacological therapy on cancer pain intensity of adults with câncer.

Overall Impression

This manuscript has relevant content for the patients and clinicians.

The manuscript is interesting, however it requires some adaptations for potential publication. I would like to make a few comments and discuss some issues that arose during the review.

ABSTRACT

-The writing is confusing and lacks an expected logical sequence. I suggest incorporating the use of PRISMA-NMA as a criterion for reporting the information in this manuscript.

- I suggest including the other objectives of the study in the abstract, such as “total effective rate of treatment”.

- Statistical model must be briefly described in the methods, in addition to which procedures will be followed during data extraction and analysis.

- I believe it is important to add the protocol registration codes.

INTRODUCTION

- “Today, the principles of three-step therapy for pain relief are often used in clinical practice”, what are they?

- “although treating cancer pain still presents difficult obstacles posed by opioid analgesic addiction and pharmacological adverse effects. Patients of late-stage cancer are less likely to become addicted to painkillers because they are less likely to acquire pharmacological resistance, requiring higher dosages to manage pain in later stages of the disease”

First, the problem of “pharmacological dependence and adverse effects” is addressed. Subsequently, the argument is broken with the information that patients in advanced stages are LESS likely to acquire resistance to drugs, which is why they need to be exposed to high doses.

Wouldn't they be MORE likely? I suggest reviewing your writing, with the aim of connecting the information and creating a logical sense in the story you want to develop. Reading is not fluid!

- “net meta-analysis” Correct term in the 28th line of page 2.

- Introduction does not support the research airms. Arguments are not coherent, making it difficult for the reader to perceive the importance of the study.

METHOD

- If it was not used, I suggest using the PRISMA-NMA extension to structure the contents of this systematic review and network meta-analysis.

Eligibility criteria

- I suggest placing the summary of the eligibility criteria (PICOTS) within the manuscript, to facilitate reader understanding.

Population

- I believe it is important to define which diagnostic criteria will be used to define the sample.

- What types of clinical trials will be included? I suggest describing the possible types of randomized clinical trials, for example parallel group, cross-over or cluster RCT.

- “The subjects of the research include those with pain that is not related to cancer”. I think the correct would be “DO NOT include”.

Interventions

- How were the intervention sets to be included pre-defined? I recommend using previously published clinical guidelines and systematic reviews as a basis. I suggest searching the literature for recently published articles, with the aim of creating reliable and relevant nodes for the condition of interest.

Comparator

- I believe it is not appropriate to incorporate the interventions into a single node: standard drug treatment, placebo and no intervention. Each intervention above controls for completely different confounding factors. Such a decision would violate the principle of transitivity, the primary criterion for conducting network meta-analysis. I suggest creating three different control groups, "placebo+sham", "no intervention+waiting list" and "standard drug treatment".

- Regarding the "standard drug treatment" group, it is necessary to be aware of the lack of consistency in its definition in clinical trials. Including different intervention modalities in the same node can potentially violate the assumption of transitivity for the NMA. I suggest creating criteria for the use of this group.

Outcomes

- Please clarify what the primary and secondary outcomes will be. It is unclear, will quality of life be a secondary outcome? I suggest placing primary outcomes and secondary outcomes in different paragraphs.Study design

-“The list will only contain items that have been published in English”. I suggest not restricting your search by language. Controlling this bias will give more credibility to potentially identified estimates.

Data extraction

- I suggest better developing the variables extracted from those eligible, separating specific information from the topics: methods, participants, interventions and outcomes. For example: Methods: study design, study setting, and year of publication; Participants: sample size, mean age, sex, disease duration, comorbidities, diagnostic criteria...

Strategy for data synthesis and analysis

- It is not clear whether the assumption of transitivity and coherence will be realized. If not programmed, I suggest adding it as a measure to ensure the validity and interpretability of the effect estimates found.

- I suggest the use of SUCRA (Surface Under the Cumulative Ranking curve), with the aim of evaluating and classifying the relative effectiveness of different treatments in relation to the outcomes of interest.

Reviewer #2: I congratulate the authors for their work, but I have a number of suggestions/concerns with this protocol. Below, I respectfully presented them to the authors:

ABSTRACT

1. Remove the sentence "offering substantial proof so that decisions can be made with understanding" from the Background. It may be in the Discussion section, not in the Background.

2. "Method" should be "Methods".

3. "in comparison to standard care, no intervention, or a placebo". * It's confusing to me. If it is a network meta-analysis, both studies with pharmacological and non-pharmacological treatments will be included, whether in comparison with another active or placebo. Is not true? I suggest that the sentence be rewritten to something like: "To be included in the review, studies must be randomized controlled trials evaluating the effects of pharmacological and non-pharmacological treatments in patients with cancer pain".

4. I suggest removing the phrase "We will follow the Preferred Reporting Items for Systematic Review and Meta-Analysis Protocols (PRISMA-P) 2015 checklist".

5. Add how the results will be presented: MD with credible intervals using a league table? Will treatments also be classified using the surface under the cumulative ranking curve (SUCRA) method? Make it clear to the reader.

6. The "Discussion" is poor. Improve, making clear its importance, given the scarcity of studies in this area, etc. Furthermore, the phrase "This network meta-analysis compares the efficacy of pharmacological versus non-pharmacological treatments for pain intensity in patients with cancer pain" should be in the future.

METHODS

1. The section must be "Methods" and not "Method".

2. In the "Population" section, remove "Inclusion criteria". Keep just "Population". The general section is already called "Eligibility criteria".

3. I suggest "in the PROSPERO" instead of "with PROSPERO".

4. "The World Health Organization reports that individuals with cancerous pain who are 18 years of age or older will be included in the study. Chronic cancer pain is defined by the International Association for Pain Research (IASP) as persistent pain caused by primary cancer lesions, metastatic lesions, or cancer-related treatments".

- I didn't understand the meaning of that sentence. In the "Population" section, describe only the eligible criteria for the population of interest. For example:

Population: Adults (18 years or older) diagnosed with cancer... and so on. Furthermore, consider informing whether the inclusion criteria are independent of gender and race. The reader cannot be left in doubt.

5. "The experimental group received medication analgesia or combination therapy (drug therapy along with non-drug therapy, conventional analgesics and unconventional analgesics); (3) The research type of literature is randomized controlled study; (4) The primary outcome measures are pain intensity score and total effective rate of treatment; (5) Secondary indicators are pain onset time, duration of relief, and quality of life score".

- This entire sentence should not be in the "Population" section. It's not the appropriate place.

6. Exclusion criteria

- Much information is unnecessary, such as "Non-randomized controlled research, including reviews, case studies, and trials on animals". The PICOs Criterion should already be clear enough. Furthermore, this section should be the last one, after "Population", "Intervention", "Comparator", "Outcomes", and "Design".

7. Comparator

- Make it clear that for network meta-analysis, due to the very nature of this study, comparisons of interventions will also be compared with each other.

8. Outcomes

- "Pain intensity was used as the primary outcome measure and measured using measurement tools". Why is the sentence in the past? This is a protocol. So, it must be in the future. Furthermore, it separates the outcomes as "primary(s)" and "secondary(s)".

9. Study design

"We divide intervention measures into two categories based on their mechanisms: pharmacological intervention and non-pharmacological intervention, also include them in joint interventions. The databases PubMed, Web of Science, the Cochrane Library, MEDLINE, Embase, and CINAHL will all be searched. The list will only contain items that have been published in English; unpublished pieces will not be included. "Cancerous pain" and "randomized controlled trials" will be included in the search phrases".

- This entire sentence should be removed from this place. Please state here only which randomized controlled trials will be included. Crossover studies will be considered in their complete form or only if there is a washout period for a few weeks? Will there be a restriction on language or publication date?

10. After the PICOs criterion, create a section "Electronic search strategies", and include information about which databases will be searched, whether they will also search gray literature, whether the authors of articles with missing data will be contacted, and whether perform hand searches for potentially eligible articles.

11. After the search section, create a "Search strategy" section, and describe the search strategies that will be used (provide as supplementary material). Describe the terms Medical Subject Headings (MeSH) and their synonyms and Boolean operators (when possible) to improve searches.

12. Create a section "Risk of bias across studies" and describe CINEMA. It shouldn't be in the section it is in.

13. Strategy for data synthesis and analysis

- Will heterogeneity be measured using the I² statistic? To estimate heterogeneity between studies, will the DerSimonian & Laird estimator be used?

- Will a number of simulations and thin values be chosen by checking autocorrelation plots and trace plots?

- If it is a Bayesian network meta-analysis, will the results be presented with credible intervals through a league table?

- "To assess the presence of publication bias or small sample effects, a corrective funnel plot was created for outcome measures having a literature volume of 10 or above." - PHRASE IN THE PAST?

14. Discussion

- "To compare the results and effects of various intervention measures, the ROB 2.0 tool was used to assess the risk of bias in the numerous studies that were included in the analysis(24)". - UNNECESSARY PHRASE.

- Highlight in the Discussion that this protocol was written guided by the PRISMA-P statement, which will endorse the transparency, accuracy, and integrity of the study.

- End the discussion with a sentence drawing attention to the lack of studies directly comparing different interventions in the scenario of cancer pain.

Additional comments:

- Review the entire text for language. As is protocol, actions must be in the future, not in the past.

- Figure 1 must be in accordance with the most recent PRISMA flowdiagram (https://www.prisma-statement.org/prisma-2020-flow-diagram).

**Do you want your identity to be public for this peer review?** For information about this choice, including consent withdrawal, please see our Privacy Policy

Reviewer #1: No

Reviewer #2: No

---

## [Author Response · Author response to Decision Letter 1]

12 Jul 2024

Dear editors and reviewers

Thank you so much offering us so many helpful, detailed, and warm-hearted comments. We learnt a lot from your advice. It's my pleasure to get your nice comments. They are all very beneficial to our team work. Although we have never met in real life, but your kind suggestions impressed me a lot. We had been rejected for many times, only you offer us a feedback, really great to meet you online. If there are any errors or mistakes, we would be very happy to corrrect them. We all really appreciate your help! Hope you all lead a happy and harmonious life and also make more progress of work.

Best regards,

Wang Yanru

---

## [Decision Letter · Decision Letter 1]

26 Mar 2025

Efficacy of Pharmacological and Non-pharmacological Therapy on Chronic Cancer Pain Intensity of Adults with Cancer: A Network Meta-analysis Protocol

PONE-D-24-07817R1

Dear Dr. Wang,

We’re pleased to inform you that your manuscript has been judged scientifically suitable for publication and will be formally accepted for publication once it meets all outstanding technical requirements.

Kind regards,

Marcus Tolentino Silva, Ph.D.

Academic Editor

PLOS ONE

Additional Editor Comments (optional):

Reviewers' comments:

Reviewer's Responses to Questions

**Comments to the Author**

1. Does the manuscript provide a valid rationale for the proposed study, with clearly identified and justified research questions?

Reviewer #2: Yes

Reviewer #3: Yes

2. Is the protocol technically sound and planned in a manner that will lead to a meaningful outcome and allow testing the stated hypotheses?

Reviewer #2: Yes

Reviewer #3: Yes

3. Is the methodology feasible and described in sufficient detail to allow the work to be replicable?

Reviewer #2: Yes

Reviewer #3: Yes

4. Have the authors described where all data underlying the findings will be made available when the study is complete?

Reviewer #2: No

Reviewer #3: Yes

5. Is the manuscript presented in an intelligible fashion and written in standard English?

Reviewer #2: Yes

Reviewer #3: Yes

You may also provide optional suggestions and comments to authors that they might find helpful in planning their study.

Reviewer #2: I congratulate the authors on the Protocol corrections and wish them luck with the meta-analysis and publication.

Reviewer #3: the study provides a holistic view of pain management, which is crucial given the side effects of opioid-based treatments.

The use of network meta-analysis (NMA) allows for direct and indirect comparisons between multiple treatments, overcoming the limitations of traditional meta-analysis.

The study strictly follows the Preferred Reporting Items for Systematic Reviews and Meta-Analyses for Network Meta-Analyses (PRISMA-NMA) guidelines.

This patient-centered approach makes the findings more applicable to real-world clinical practice.

While the study considers adverse effects , it does not specify how they will be compared across interventions. this is the only issue i feel which is not properly addressed.

**Do you want your identity to be public for this peer review?** For information about this choice, including consent withdrawal, please see our Privacy Policy

Reviewer #2: **Yes: ** Filipe Ferrari

Reviewer #3: No

---

## [Editor Report · Acceptance letter]

PONE-D-24-07817R1

PLOS ONE

Dear Dr. Wang,

I'm pleased to inform you that your manuscript has been deemed suitable for publication in PLOS ONE. Congratulations! Your manuscript is now being handed over to our production team.

Kind regards,

on behalf of

Prof Marcus Tolentino Silva

Academic Editor

PLOS ONE